# Comparison of Imitation Crab Sticks with Real Snow Crab (*Chionoecetes opilio*) Leg Meat Based on Physicochemical and Sensory Characteristics

**DOI:** 10.3390/foods11101381

**Published:** 2022-05-10

**Authors:** Sohyun Mun, Eui-Cheol Shin, Seonghui Kim, Joodong Park, Chungeun Jeong, Chang-Guk Boo, Daeung Yu, Jin-Ha Sim, Cheong-Il Ji, Taek-Jeong Nam, Suengmok Cho

**Affiliations:** 1Department of Food Science and Technology, Institute of Food Science, Pukyong National University, Busan 48513, Korea; ansth23@gmail.com (S.M.); shkim.pknu@gmail.com (S.K.); jeongpknu@gmail.com (C.J.); 2Department of GreenBio Science/Food Science, Gyeongsang National University, Jinju 52725, Korea; eshin@gnu.ac.kr (E.-C.S.); dbs7987@naver.com (C.-G.B.); 3Guru Partners Co., Ltd., Jeju 63597, Korea; joodong.park@gurupartners.co.kr; 4Department of Human Senior Ecology Cooperative Course (Food and Nutrition), Changwon National University, Changwon 51140, Korea; duyu@changwon.ac.kr (D.Y.); wlsgk3272@naver.com (J.-H.S.); 5Department of Food and Nutrition, Changwon National University, Changwon 51140, Korea; 6Lucky Union Foods Co., Ltd., Samutsakorn 74000, Thailand; cheongilji@gmail.com; 7Future Fisheries Food Research Center, Institute of Fisheries Sciences, Pukyong National University, Busan 46041, Korea; namtj@pknu.ac.kr

**Keywords:** imitation crab stick, crab-flavored surimi seafood, snow crab meat, similarity, physicochemical properties, sensory evaluation

## Abstract

Recently, many manufacturers have been developing or producing imitation crab sticks (ICSs) that are highly similar to real snow crab leg meat (RC). This study evaluated the similarities between commercial ICSs and RC based on the analysis of physicochemical and sensory properties. Normal ICS (NS) and premium ICSs either with real crab leg meat (PS-RC) or without it (PS) were compared with RC. The sensory evaluation results showed that PS and NS had the highest and lowest levels of similarity to RC, respectively. The carbohydrate contents of ICSs (10–23%) were higher than that of RC (0.5%). Among ICSs, PS showed more similarity with RC than NS and PS-RC in terms of gel strength and texture profiles. PS-RC and PS showed a microstructural pattern that slightly imitated the muscle fiber arrangement of RC. The electric tongue analysis of taste compounds, such as sugars, free amino acids, and nucleotides, showed that the taste profile of ICSs is distinctly different from that of RC. The electronic nose analysis identified 32 volatile compounds, while the principal component analysis using electronic nose data successfully distinguished three clusters: PS-RC and PS, RC, and NS. Our results could provide useful information for the development of ICSs with higher similarity to RC.

## 1. Introduction

The imitation crab stick (ICS), also known as crab-flavored surimi seafood, is a well-known imitation food worldwide [1]. It is a mostly surimi-based product that derives its elasticity from surimi gel formation. ICSs also contain starch, flavorings, and colorants [2]. Commercial ICSs are manufactured using automatic processing machinery. Briefly, surimi mixed with other ingredients is extruded as a thin sheet and then steamed to create surimi gel. The steamed surimi sheet is then bundled and rolled into a rope shape to imitate the appearance and texture of real snow crab leg meat (RC) [3].

The ICS was invented in 1975 in Japan to develop an alternative to high-cost RC. It has since been commercialized in Asian countries [4] and widely accepted in Western countries, including the US and Europe, since the early 1980s [5]. ICSs have been produced in various sizes and types (flake type and stick type), depending on different cultures of food consumption or intended purpose [6]. Generally, commercial ICSs can be divided into normal and premium products. In particular, the premium ICSs are produced closer to the real crab meat via the addition of real crab meat or processing technique.

To date, most studies on ICSs have focused on improving manufacturing methods and adding various ingredients to improve their quality, including their gelling and sensory properties [7]. Other studies have reported the preparation of ICSs using terrestrial meats such as chicken, pork, and beef [2]. This study aims to promote the development of high-quality ICS products with high similarity to RC. To the best of our knowledge, no previous reports have quantitatively compared the quality characteristics of ICSs and RC.

Our study evaluated the similarities between commercial ICSs (normal and premium products) and RC based on the analysis of physicochemical and sensory properties. Similarities between ICSs and RC were evaluated by comparing their chemical compositions, gel strengths, texture profiles, microstructures, and color values, and by analyzing their smell and taste profiles using an electronic nose and tongue.

## 2. Materials and Methods

### 2.1. Materials

ICSs were purchased from a local market (Busan, Korea). Commercial ICSs were classified by the type and content of raw materials as follows: normal ICS (NS), premium ICSs with red snow crab (*Chionoecetes japonicus*) leg meat (PS-RC), and premium ICSs without red snow crab leg meat (PS) (Figure 1A). According to product information supplied by the manufacturer, the PS-RC contains 6% red snow crab leg meat. Both PS-RC and PS contain artificial crab flavoring. Frozen snow crab (*Chionoecetes opilio*) leg meat (RC) was purchased from Mag-Sea International Co., Ltd. (Magadan, Russia). ICSs and RC were stored at 5 °C and −80 °C, respectively. Before experiments, RC was boiled for 10 min after defrosting at room temperature (20–25 °C) for 30 min. All chemicals and reagents used in this study were of analytical grade. All standard samples and eluents were purchased from Sigma (Sigma-Aldrich Inc., St. Louis, MO, USA) and were of high-performance liquid chromatography (HPLC) grade.

### 2.2. Physical Properties

#### 2.2.1. Gel Strength

The gel strength was measured using a rheometer (CR-100D, Sun Scientific Co., Ltd., Tokyo, Japan) under the following conditions: MODE 20; adapter type: cylindrical plunger (diameter: 5 mm); penetration speed: 60 mm/min; load cell (max): 2 kg; pressure distance: 50%. The samples were cut into 1 cm^3^ pieces and equilibrated for 30 min at room temperature prior to measurement. Ten individually prepared samples were tested.

#### 2.2.2. Texture Profile

A texture profile analysis was performed using a texture analyzer (CT3 4500, Brookfield Engineering Laboratories Inc., Middleboro, MA, USA) with a cylindrical plunger (diameter: 12.7 mm). The sample preparation process was the same as that for measuring gel strength. The samples were compressed to 50% of their original height with a 2 kg load cell at a speed of 1 mm/s and subjected to a two-cycle compression test. The hardness, springiness, gumminess, chewiness, and cohesiveness were measured. Ten individually prepared samples were tested.

#### 2.2.3. Color

The Hunter color values of ICSs and RC were measured using a colorimeter (SP60, Lovibond Co., Amesbury, UK). Samples were cut into cubes (1 cm thickness, 1 cm length, 1 cm height) and then cut vertically. The surfaces of the cut samples were used to measure for lightness (*L**), redness (*a**), and yellowness (*b**), while the insides of the cut samples were used to measure whiteness. All samples were measured three times. Whiteness was determined using the following equation:(1)$$\mathrm{whiteness}=100-\sqrt{{\left(100-L\right)}^{2}+a^{2}+b^{2},}$$

#### 2.2.4. Scanning Electron Microscope (SEM)

Samples were cut into pieces (5 mm thickness) and fixed with 2.5% glutaraldehyde in 0.2 M phosphate buffer for 2 h at room temperature. The specimens were washed in deionized distilled water three times for 15 min each time. Then, the specimens were washed in 50% ethanol (*v*/*v*) two times for 5 min, in 70% ethanol for 10 min, in 80% ethanol for 10 min, in 90% ethanol for 10 min, and in100% ethanol two times for 20 min. All specimens were coated with gold using an ion sputter (E-1010, Hitachi High Technologies Co., Tokyo, Japan) and observed using a low-vacuum scanning electron microscope (LV-SEM) operating at 15 kV.

#### 2.2.5. Inverted Microscope

Samples were placed in 15% sucrose in distilled water at 4 °C for 1 h. The samples were then covered with an optimum cutting temperature compound. They were quickly frozen and 30 µm sections were cut using a cryostat (Microm HM 525 NX, Thermo Scientific, Waltham, MA, USA) set at −30 °C. Each specimen was mounted on a microscope slide. The microstructures of the prepared specimens were observed using an inverted fluorescence microscope (IX51-A12PH, Olympus Co., Tokyo, Japan).

### 2.3. Chemical Properties

#### 2.3.1. Proximate Composition

The proximate compositions were determined using standard analytical methods (AOAC, 1990) [8]. The moisture content was determined using the oven-drying method. The amount of crude protein in each sample was determined using the micro-Kjeldahl method, the lipid content was determined via Soxhlet extraction, and the ash content was determined using the dry-ashing method. The total carbohydrate content was calculated as follows: 100 − (moisture content + lipid content + ash content + protein content). All analyses were performed in triplicate.

#### 2.3.2. Sugars

Five grams of each sample was homogenized and placed in a separate polyethylene tube containing 25 mL of petroleum ether and centrifuged at 8000× *g* for 10 min. The petroleum ether was then removed under a stream of nitrogen gas. Lipids were extracted from the samples by adding 25 mL of 50% (*v*/*v*) ethanol in a water bath at 85 °C for 25 min. The samples were then centrifuged at 2000 rpm for 10 min and filtered through a 0.45 μm membrane filter. Sugars were analyzed via HPLC (Agilent 1100 Series, Agilent Technologies, Santa Clara, CA, USA). The HPLC system was equipped with a refractive index detector. All analyses were performed in triplicate. The injection volume was 10 μL. The column used for separation was a μBondapak C18 (300 mm × 3.9 mm, 10 μm, Waters, Milford, MA, USA). The column temperature was maintained at 30 °C. The mobile phase consisted of acetonitrile and water (80:20 *w*/*w*). The flow rate was 1.0 mL/min.

#### 2.3.3. Free Amino Acids

Trichloroacetate (TCA) extract was used in the analysis of amino acids. Three grams of each sample was homogenized twice for 15 min at ambient temperature in 3 mL of 16% (*v*/*v*) TCA. Homogenized samples were then centrifuged at 3000 rpm for 15 min. The supernatants from the first and second extractions were combined and filtered using a 0.22 μm filter. Free amino acid analysis was performed using an automatic amino acid analyzer (L-8900, Hitachi High Technologies Co.). All analyses were performed in triplicate.

#### 2.3.4. Nucleotides

Nucleotides were identified according to the method described by Chen and Zhang (2007). Samples were minced, and then 5 g of each sample was homogenized in 25 mL of 0.6 M perchloric acid for 2 min before being centrifuged at 8000× *g* for 10 min. The supernatant was filtered and neutralized with 1 M potassium hydroxide (KOH). After neutralization (pH 6.5–7.0), the solutions were equilibrated at 0 °C for 30 min and filtered through 0.2 μm filters. Deionized water was used to adjust all sample solutions to a volume of 20 mL before they were put through 0.2 μm filters. Nucleotides were analyzed via HPLC (Agilent 1100 Series, Agilent Technologies). The HPLC system was equipped with an ultraviolet detector. An Eclipse XDB C18 column (150 mm × 4.6 mm, 5 μm, Agilent) was used for separation. The column temperature was maintained at 30 °C. The eluents used were 1% triethylamine and 1% phosphoric acid (pH 6.5; *w*/*w*). The flow rate was 1 mL/min. The detection wavelength was set at 254 nm. The injection volume was 10 μL. All analyses were performed in triplicate. The identity and quantity of the nucleotides were determined by comparison with the retention times and peak areas of each nucleotide standard.

#### 2.3.5. Electronic Tongue

Taste intensity patterns from each sample were analyzed using sensors that detect individual flavor compounds and an electronic tongue system (ASTREE, Alpha MOS, Toulouse, France) with an Ag/AgCl reference electrode. SRS, STS, UMS, SWS, and BRS were used to detect sourness, saltiness, savoriness, sweetness, and bitterness, respectively, whereas GPS and SPS were used as reference sensors to correct the other sensor values. First, 10 g of each sample was stirred at 150 rpm for 60 min at 50 °C and in 100 mL of purified water to extract the flavor compounds. Fine particles that could affect the analysis and other solids were removed via filtration. Subsequently, 100 mL of each sample solution was loaded into the sampler of the electronic tongue and kept in contact with the sensor for 120 s. Taste patterns were examined by analyzing each sample five times, and flavor compound patterns between samples were examined [9].

#### 2.3.6. Solid-Phase Microextraction (SPME)–Gas Chromatography (GC)–Mass Spectrometry (MS) Analysis and Identification of Aromatic Compounds Using GC–Olfactometry Testing

The headspace analysis method, which uses SPME fibers (Supelco Co., Bellefonte, PA, USA) coated with 100 μm of polydimethylsiloxane, was used to collect the volatile compounds in each sample. Five grams of the sample was placed into the odor collection bottle, and before sealing it with an aluminum cap the fibers were exposed to sample vapor at 50 °C for 30 min to induce the absorption of volatile compounds. Volatile compounds were analyzed via GC-MS (Agilent 7890A and 5975C, Agilent Technologies) with an HP-5MS column (30 m × 0.25 mm i.d., 0.25 μm film thickness). The oven temperature program was held for 5 min at 40 °C and then heated to 200 °C at 5 °C/min, and the injector temperature was set to 220 °C so as to segregate volatile compounds. Each compound was identified and quantified using the National Institute of Standards and Technology (NIST) library (ver. 12) and an internal standard (C15:0). The GC-MS results of each sample were subjected to PCA to study the patterns of the volatile aromatic compounds [9].

Each volatile compound segregated by GC-MS was subjected to a sniffing test using an olfactory detection port with a heated mixing chamber (ODP 3, Gerstel, Inc., Linthicum, MD, USA) attached to the GC-MS detection system. Given how olfactory senses vary between individuals and how olfactory sensitivity decreases over time, three skilled experimenters participated in the same experiments, and their reactions to the strength of each volatile compound were recorded.

#### 2.3.7. Electronic Nose

To analyze volatile compounds, 3 g of each sample was prepared in a headspace vial and stirred for 20 min at 50 °C to collect headspace compounds. After collecting 5000 μL of volatile compounds using an automatic sampler and injecting them into the electronic nose, two columns (MXT-5/MXT-1701) were installed in parallel and the components were analyzed using two flame ionization detectors. The acquisition time was 230 s, and the analysis was performed with a trap absorption temperature of 40 °C and a trap desorption temperature of 240 °C. Compounds were identified based on their correlation with segregated peaks using AroChemBase (Alpha MOS), which is based on Kovat’s index library, which includes approximately 88,000 compounds. Each sample was analyzed five times, and discriminant patterns among the samples were examined [10].

### 2.4. Sensory Evaluation

The difference-from-control test was used to determine how different the sample was from the control. A sensory evaluation was performed by a panel of 10 trained panelists (5 males and 5 females, 22–27 years old). RC was labeled “control”, and other samples were labeled with a coded number. Room-temperature test samples were served to the panelists on a white plate. The panelists were requested to assess each coded sample for overall quality (color, texture, aroma, taste, and overall acceptance) compared with the control using a 9-point scale (1 = not different, 9 = extremely different).

### 2.5. Statistical Analysis

All results are expressed as the mean ± standard deviation (SD) of independent measurements. Here, *p*-values of less than 0.05 were considered significant for all statistical tests. For performing multiple comparisons, the data were analyzed using one-way analysis of variance followed by Dunnett’s test. Data analysis was performed using GraphPad Prism 5.0 (GraphPad Software, Inc., San Diego, CA, USA). The significance of differences between mean values was determined by one-way analysis of variance (ANOVA). A multivariate analysis of variance, followed by Duncan’s multiple comparison test (MANOVA), was performed in proximate composition data. Data analysis was performed using SPSS 11.0 software (SPSS Inc., Chicago, IL, USA). Electronic sensor data, tastes, and flavors were patterned and clustered using principal component analysis, one of the multivariate analyses. The PCA was conducted using XLSTAT software ver. 9.2 (Addinsoft, Paris, France) to identify how samples (objects) and sensory responses (variables) were located in the chemosensory characteristic patterns. The PCA plots (score and loading plots) in the study were based on Pearson’s correlation. Plots produce a set of new orthogonal axes or variables known as principal components (PCs) from the original variables. The resulting sets presented on the orthogonal axes explain variance in the data set through the comparison of only a few PCs.

## 3. Results and Discussion

### 3.1. Sensory Evaluation

Sample images may be helpful in understanding the results of sensory evaluations and color values. The ICSs and RC used in this study are shown in Figure 1A. The difference-from-control test was adopted to evaluate the similarities between ICSs and RC. This test was designed to evaluate whether differences exist between one or more samples and a provided control and to estimate the degree of those differences [11]. The panelists were requested to evaluate each coded sample (commercial ICSs) for overall quality (color, texture, aroma, taste, and overall acceptance) compared with the control (RC) using a 9-point scale (1 = not different, 9 = extremely different). As a result of the overall acceptance of the participants in the sensory evaluation (Figure 1B), we found that premium ICSs (PS-RC and PS) were more similar to RC than to NS. In particular, NS was considerably different from RC for all sensory evaluation items. The sensory evaluation, except for overall acceptance, was explained with matched physicochemical properties.

### 3.2. Proximate Composition

RC consisted of 78.9% moisture, 17.8% protein, 0.5% carbohydrate, 1.2% lipid, and 1.6% ash (Table 1). This proximate composition of RC was similar to the result from a previous report (78.6% moisture, 18.5% crude protein, 0.3% carbohydrate, 1.2% lipid, and 1.4% ash) [12]. In contrast to RC, all ICS samples contained a large amount of carbohydrate (10–23%). This is because starch is added to surimi to improve the gel strength and yield of ICS production [2]. The carbohydrate contents of PS-RC (11.1%) and PS (10.7%) were significantly (*p* < 0.05) lower than that of NS (23.3%). Generally, for price competition, more starch is added to NS than to PS-RC or PS. The addition of carbohydrate ingredients to ICS production results in decreased protein content. The protein contents (6–11%) of all ICSs were significantly lower than that of RC (17.8%). Since NS had a high carbohydrate content, it had a lower protein content (6.3%) than PS-RC (11.1%) and PS (10.7%).

### 3.3. Colors

The surface (redness, *a**) and inside (whiteness) colors of ICSs are important sensory indicators that determine their similarity to RC and their commercial value. PS-RC (24.6), PS (17.6), and NS (31.7) showed significantly higher *a** values than RC (10.8) (*p* < 0.05) (Table 2). In particular, the degree of redness in the NS group was excessively high. Differences in redness between the samples were also observed in the sensory evaluation of redness (Figure 1B). ICS manufacturers have used a mixture of red colorants, such as *Monascus* pigment, cochineal extract, and paprika oleoresin, to imitate the red color of RC (Park, 2008). The overrepresented redness of NS may originate from the excessive use of pigments or the composition of red pigments representing exaggerated color.

RC (78.9) showed significantly higher whiteness values than ICSs (*p* < 0.05). PS-RC (71.1) and PS (66.5) exhibited high whiteness values. NS had the lowest whiteness value (58.5). The whiteness values of ICSs were negatively correlated with the concentration of starch. It has been reported that whiteness decreases as the concentration of starch increases in surimi products [13].

In addition, the grade and fish species of surimi also affect the whiteness values (Park and Lanier, 2000). Premium products are made from high-quality surimi, primarily Alaska pollock (*Theragra chalcogramma*) surimi [4]. Alaska pollock surimi is generally brighter in color than surimi from other species [14]. Therefore, increasing redness and decreasing starch concentration should be considered for the preparation of ICSs with higher similarity to RC.

### 3.4. Rheological Properties

Rheological properties such as gel strength and elasticity are the most important indicators of ICS quality [15]. Figure 2 shows the gel strength and texture profiles of RC and ICSs. The gel strength value of RC was 669 g∙mm, and among the ICSs, PS (774 g∙mm) showed the most similar gel strength to RC. The other premium ICS, PS-RC (480 g∙mm), had a lower gel strength value than RC and PS. The addition of red snow crab leg meat may explain the differences in gel strength between PS-RC and PS. Liang et al. [15] reported that sarcoplasmic proteins in crab meat coprecipitate with myofibrillar proteins in surimi, which decreases surimi gelation.

Among the ICSs, NS showed the highest gel strength (1223 g∙mm). The high starch content (23.3%) and low moisture content (67.4 %) of NS can be explained by it having a higher gel strength than PS-RC and PS. Generally, starch is added to surimi-based products to increase the gel strength [13]. Luo et al. [16] reported that an increase in water content decreased the gel-forming ability of surimi owing to the lower myofibril protein concentrations and decreased cross-link density.

It is well known that gel strength is correlated with hardness in surimi-based products, including ICSs [17]. Differences in hardness between RC and ICSs showed a similar trend to gel strength differences (Figure 2B). Park et al. [14] reported that the hardness of surimi-based products increases with increasing starch content. Among the ICSs, NS (1.5 kg) had the highest hardness value, which can be explained by its high starch content. PS-RC (0.3 kg) had a hardness value (*p* < 0.001) similar to that of RC (0.4 kg). The hardness of PS (0.8 kg) was higher than that of PS-RC. This difference can be explained by differences in microstructure. A gel structure with a larger void size results in low hardness in surimi-based products [18]. In this study, PS-RC exhibited a gel structure with larger voids (Figure 3) than PS. According to Liang et al. [15], the addition of red snow crab leg meat increases the void size in surimi-based products [15].

Chewiness and gumminess are calculated based on hardness. Increases in both textural parameters were in line with the increase in hardness [19]. In the present study, chewiness (Figure 2C) and gumminess (Figure 2D) correlated positively with hardness, as in [19]. Springiness is described as how rubbery the gel feels in the mouth and indicates how well the sample physically recovers its original height after deformation [19]. NS (4.2 mm) and PS (3.8 mm) showed significantly higher springiness values than RC (3.1 mm) and PS-RC (3.1 mm). The high springiness of NS and PS can be explained by their hardness values; it is well known that a high springiness value correlates with a high hardness value [20].

Cohesiveness is an important index of the quality of surimi-based products and is the most sensitive parameter that expresses gel-forming ability [21]. Among the samples, the cohesiveness of PS-RC (0.3) was significantly lower than that of RC (0.5) and the other ICSs (0.5) (Figure 2F) (*p* < 0.001). The lower cohesiveness of PS-RC can be explained by the poor gel-forming ability of PS-RC due to the addition of red snow leg crab meat. PS-RC had the lowest gel strength and largest voids (Figure 3). Generally, a good gel-forming ability of surimi results in high cohesiveness and a denser microstructure [22]. Differences in rheological properties between RC and ICSs were observed in the sensory evaluation of the texture item (Figure 1B). These results match with our observations about differences in gel strength between RC and ICSs.

### 3.5. Microstructure

To better understand the physical properties of RC and ICSs, their microstructures were observed and compared using inverted microscopy and SEM (Figure 3). In the inverted micrographs (×20), the arrangement and gaps of muscle fibers of RC were observed. Microstructural patterns observed with the inverted microscope were confirmed with the SEM micrographs (×50). The RC muscle bundle contained an average of 152 strands (*n* = 10), and its muscle fibers had variable diameters with a range of 427–1046 μm (mean diameter: 676 μm). The difference in diameter between muscle fibers in the RC was also observed in the report by Perry et al. [23].

PS-RC and PS had an irregular gel structure containing voids and showed a microstructural pattern that slightly imitated the arrangement of RC muscle fibers. These microstructural patterns of ICSs are artificially formed in the manufacturing process, which bundles and rolls thin surimi sheets [3]. The gel structure of PS-RC had larger voids than those of PS. Different rolling and compression packaging processes may cause differences in the microstructural patterns and void sizes of PS-RC and PS. A smaller void size implies a more compact gel network structure, resulting in a higher gel strength and hardness [18]. The differences in void size between premium ICSs were explained by the fact that PS-RC showed a lower gel strength and hardness than PS. In addition, the addition of red snow crab leg meat may explain the differences in void size between PS-RC and PS. Liang et al. [15] reported that the void size increased with an increase in the amount of added crab meat.

NS did not show a microstructural pattern that imitated the arrangement of RC muscle fibers. Among the samples, NS had the most compact structure with few voids. This structure can be explained by the high starch content of NS. Mi et al. [24] reported that the addition of starch led to a more compact microstructure and increased gel strength. At a higher magnification (×500), larger cavities were observed in PS but not in PS-RC. The surimi grade can explain this difference. Wei et al. [25] reported that high-grade surimi forms a more compact and denser microstructure with smaller cavities.

### 3.6. Taste

To compare the tastes of ICSs and RC, sensory evaluations and measurements of taste compounds (sugars, free amino acids, and nucleotides) were performed. In addition, the taste profiles of the ICSs and RC were analyzed using an electronic tongue. With regard to the sensory evaluation of taste items (Figure 1B), NS was considerably different from RC. Premium ICSs (PS-RC and PS) were more similar to RC than to NS. In particular, it was shown that PS was most similar to RC. As a result of the taste item (Figure 1B), unlike our prediction, the taste of ICSs was not affected by the presence of red snow crab leg meat. The taste of ICSs and their similarity to RC would be sufficiently increased by the addition of crab extract. Therefore, the addition of a sufficient amount of crab extract, which is more cost-effective than adding real crab meat, should be considered to improve the taste of ICSs.

Among the sugar compounds, only sucrose (2.4–4.4 g/100 g) was found in ICSs, and RC did not contain any sugars (Table 3). This is because sucrose is added as a cryoprotectant to surimi before freezing [14].

Nucleotides are important factors in umami taste [26]. Inosine monophosphate (IMP) and guanosine monophosphate (GMP) are intense flavor enhancers for umami taste and are much stronger than monosodium glutamate (MSG) [26]. Konosu et al. [12] reported that snow crabs contain high amounts of adenosine monophosphate (AMP) and small amounts of IMP and GMP. Unlike previous results, IMP and GMP were not identified in our study (Table 3). This result can be explained by the boiling treatment of frozen snow crabs. During boiling, the AMP in the snow crab increases, whereas the other nucleotides greatly decrease [27]. The ICSs contained only IMP and GMP and not AMP. In addition, the IMP content of ISCs was higher than that of GMP. Alaska pollock (*T. chalcogramma*) surimi is the main ingredient in commercial ICSs. It contained a high amount of IMP and a low amount of AMP [28]. The taste of PS was more similar to that of RC than PS-RC in the sensory evaluation (Figure 1B). PS-RC had higher IMP and GMP contents than PS. As a result of the taste, we found that the increases in IMP and GMP were not related to the increase in similarity to RC.

Amino acids contribute to sour, bitter, and sweet tastes [26]. Glycine (sweet), alanine (sweet), proline (sweet/bitter), and arginine (bitter/sweet) were the major free amino acids in RC (Table 4). The composition of free amino acids in RC (Table 4) was similar to that in a previous report [12] and was different from that in ICSs. Among the free amino acids, arginine showed the largest difference. The arginine content (498 mg/100 g) of RC was significantly higher than that of ICSs (0.8–1.6 mg/100 g). Arginine (bitter/sweet) contributes to a pleasant overall preference when eating crab [26], and Alaska pollock has a low arginine content [28]. ICSs had higher contents (189–257 mg/100 g) of glutamic acid than RC (24.2 mg/100 g). Glutamic acid has a sour taste and contributes to umami taste in the presence of sodium salt [26]. Similar to snow crab leg meat, Alaska pollock had a low amount of glutamic acid [27,28]. The high glutamic acid content of ICSs can be explained by the addition of MSG to the crab flavoring. When MSG is added to food, it is detected as glutamic acid [29].

The taste patterns of each sample detected using an electronic tongue are shown in Figure 4. The RC samples analyzed using the electronic tongue sensor showed the highest umami sensor value (8.7), while the salty taste (STS) was the lowest (5.4). NS showed relatively high saltiness and low umami (4.5), sweetness (5.8), and bitterness (5.5) values. Premium-grade samples (PS-RC and PS) showed relatively higher bitterness (6.6 and 6.3), but otherwise showed no notable difference in flavor compound intensity. The testing of taste compounds involves sensory tests with skilled panelists, but differences in personal preferences, culture, gender, and age can still influence the results. Therefore, electronic tongues using electronic sensors are actively used in the quality control of food and food materials. Although they cannot accurately replicate the complex and delicate human taste receptor system, they can provide objective information regarding each sample [9]. An electronic tongue is a device that patterns the taste of a sample by measuring the relative response value of the sensor for a compound of interest and comparing it to that of a compound that represents the taste of the sample. Lee et al. [30] reported a pattern of taste compounds depending on the extraction stage of ginseng using an electronic tongue, while Xiao et al. [31] determined the grades for various teas based on their flavor compounds. Research regarding the origins of food materials [32] is becoming increasingly important, and it is believed that standardization using an electronic tongue will be beneficial in such studies.

Segregation between the samples was verified via PCA, and the results are shown in Figure 4. PC1 (principal component 1) and PC2 (principal component 2) plots explain 99.5% of the total variability in the data set; PC1 was composed of positive loads of SWS, UMS, and SRS, as well as natural loads of the other tests. Segregation of the real crab meat (RC) sample from the imitation samples was confirmed using PCA. In addition, PC2 was positively correlated with BRS (bitterness) and negatively correlated with STS, confirming the segregation from premium (PS-RC and PS) and normal impression crab stick (NS) samples.

### 3.7. Aroma

The volatile characteristics of each sample were analyzed using an electronic nose, and the results are presented in Table 5. A total of 25 types of compounds were identified using two types of analysis columns. Based on the peak area, the NS sample exhibited the highest aromatic compound content, whereas the RC sample exhibited the lowest content. The main aromatic compounds in the RC samples were ethanol (alcoholic, pungent, and sweet), 2-propanol (alcoholic, pure), and 2-methylfuran (burnt, chocolate, and glassy). The main aromatic compounds in the NS samples were ethanol (alcoholic, pungent, sweet) and 2-propanol (alcoholic, pleasant), accounting for 99% of all aromatic compounds. Ethanol was the dominant aromatic compound (315.83 ± 23.47) in all samples. The premium samples (PS-RC and PS) shared an aromatic profile distinct from the other two samples. The main aromatic compounds were identified as ethanol (alcoholic, pungent, and sweet), methyl formate (fruity, plum), pentane (alkane, gasoline), 2-propanol (alcoholic, pleasant), 2-methylfuran (burnt, chocolate, glassy), and methylcyclopentane (gasoline). Among the volatile compounds in the premium samples (PS-RC and PS), ethanol was the most abundant at 90.41% ± 8.61% and 95.81% ± 1.88%, respectively.

The degrees of segregation among sample aromatic compounds detected and quantified using an electronic nose were examined via PCA, as shown in Figure 5. The PC1 and PC2 plots explained 85.17% of total variation in the data set, and the PC1 was composed of positive loads of most volatile compounds, except for atraton, 2-methylfuran, 1-chloropentane, perfluorononane, methyl formate, methylcyclopentane, and pyridine. PC2 was positive correlated with volatile compounds except for 2,4-heptadienal, 2-pentyl pyridine, α-selinene, 8-methyl pentadecane, α-terpinen-7-al, and methyl tetradecanoate. The PC1 was composed of NS-positive and NS-negative loadings from the other components. The premium samples (PS-RC and PS) were positively correlated with the PC2 plot, whereas the RC samples were negatively correlated. Based on the PCA plot, three clusters, RC (cluster 1), premium (PS-RC and PS; cluster 2), and NS (cluster 3), were segregated successfully.

Since the electronic nose analysis system analyzes the electrochemical properties of complex mixtures of compounds with numerous aromatic compounds through selective detection using multiple sensor arrays, it is limited when compared with GC-MS, which quantifies individual aromatic compounds. However, it is convenient for determining the similarities between samples by analyzing the patterns of groups of aromatic compounds. In addition, the electronic nose system is capable of rapidly analyzing large samples and is widely used for the quality evaluation and management of food and food materials, since it can detect the odor of the entire sample simultaneously [9,10].

The volatile aromatic compounds in each sample were examined via SPME-GC-MS and GC-olfactory analyses, as shown in Table 6. A total of 14 types of odor compounds were identified, including 7 types of hydrocarbons, 4 types of acids and esters, 2 types of ketones, and 1 miscellaneous compound, respectively. The effect of hydrocarbons, which showed the highest number of aromatic compound types, on the sample aroma was very low due to their high threshold values, consistent with a previous report [33]. In terms of the aromatic compound content, the NS samples showed the highest content at 7.51 ± 4.34 g/100 g, while the RS samples showed the second highest content. No aromatic compounds that were common to all samples were identified. For the RC samples, aromatic compounds were examined via GC–olfactometry testing, and the crab meat aroma was determined from 2,3,4-trimethyl-1-butene at a retention time of 25.31 min. In addition, this compound was identified as a unique property of the real crab meat that did not appear in the other samples, and a fishy odor that was not found in other samples was identified as trimethylamine, which is a typical fishy aromatic compound. The fact that trimethylamine, a representative fishy odor compound, did not appear in the imitation crab sample demonstrates that it can be used as an odor indicator to confirm the presence of real crab meat. Cha et al. [34] also found the highest content of trimethylamine among various aromatic compounds in crabs. As trimethylamine has a very low threshold value, its detection as a volatile compound in crabs and other seafood is very important. In addition, ketone-based substances were detected only in the premium samples. According to the literature, substances in the ketone family are fatty acid decomposition products that contribute to the sweet scent of flowers or fruit flavors [34]. In the NS samples, which had a high content of aromatic compounds, all compounds except ethyl octanoate showed lower olfactory strengths than their contents, as indicated by the low odor intensity. Ethyl octanoate is an aromatic compound that has been identified in imitation samples but not in RC samples. Because the odor image is recognized as a crab stick, it can be used as another odor indicator to distinguish between real and imitation crab meats.

Aromatic compounds generated by heating king crabs [35] and the by-products of red crab processing [36] were analyzed. These studies found that aldehydes, ketones, and sulfur-containing compounds are the major components with different types and numbers of aromatic compounds, depending on the analysis method. Cha et al. [33] showed that a pyrazine-based substance, among various aromatic compounds from concentrates, is the major aromatic compound in crustaceans because of its low olfactory detection threshold. However, for the real crab meat analyzed in this study, this pyrazine-based substance was not detected because the samples were unheated. Ahn et al. [1] and Cha et al. [36] showed that the main aromatic compounds in various fish and crustaceans were sulfur-containing compounds and trimethylamine. These compounds have been reported to be important aromatic compounds because of their low olfactory detection thresholds. In addition, Chung [37] analyzed the aromatic compounds of crab meat (*Charybdis feriatus*) obtained from various locations, and trimethylamine was found to have the highest content among the individual aromatic compounds in every sample tested.

An examination of the segregation between the samples via PCA is shown in Figure 5. The PCA plot distinguished three clusters: RC (cluster 1), NS (cluster 2), and premium (PS-RC, PS: cluster 3) samples. PC 1 accounted for 44.40% of the variation in the PCA plot, with RC samples showing a positive correlation, while NS, PS, and PS-RC samples showed negative correlations. For PC2, the RC and NS samples showed a positive correlation, whereas the two premium samples (PS and PS-RC) showed negative correlations, confirming a clear segregation between the sample types. These results indicate that imitation crab products and real crab meat can be distinguished by examining their aromatic compounds.

## 4. Conclusions

Imitation crab sticks, which imitate the taste and texture of snow crab leg meat, are well known worldwide. Although 45 years have passed since ICSs were invented, studies to improve their similarity to RC continue. Nevertheless, there was no similarity comparison between ICSs and RS. We evaluated the similarities between commercial ICSs and RC based on the analysis of physicochemical and sensory properties. Our results showed that even for premium ICSs, there was a difference in similarity to RC. To the best of our knowledge, this study is the first report and could provide useful information for ICS technology. Based on the results of our study, we propose that the following objectives be pursued in order to develop ISCs with higher similarity to RC: (1) better imitation of the natural red color; (2) starch reduction to increase the whiteness and protein content; (3) modification or improvement of the crab flavoring to better approximate real snow crab flavor; (4) development of a novel bundling and rolling process to more closely approximate the microstructural and rheological properties of real snow crab meat. This study has a limitation in that only commercial ICSs distributed in Korea were evaluated. Therefore, further studies are needed to evaluate various global ICSs.

## Figures and Tables

**Figure 1 foods-11-01381-f001:**
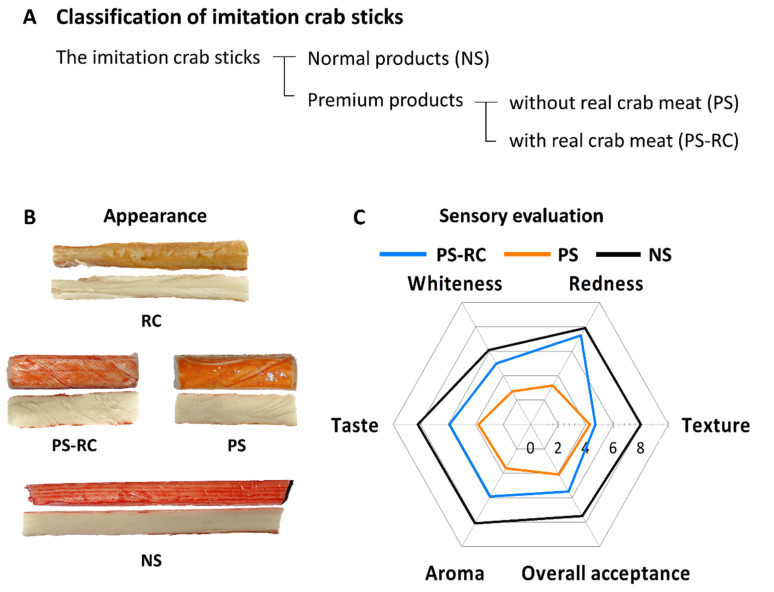
(**A**) Classification of imitation crab sticks (ICSs). (**B**) Appearance of real snow crab (*C. opilio*) leg meat (RC) and commercial ICSs. (**C**) Radar plot for the sensory evaluation of ICSs compared to RC. Scale: 1 = not different from RC, 9 = extremely different from RC. PS-RC, premium ICS with real red snow crab (*C. japonicus*) leg meat; PS, premium ICS without real red snow crab leg meat; NS, normal ICS.

**Figure 2 foods-11-01381-f002:**
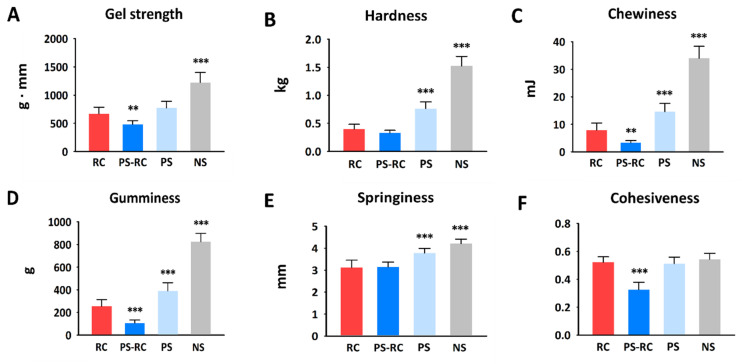
Gel strength (**A**) and texture profiles (**B**–**F**) of real snow crab (*C. opilio*) leg meat (RC) and commercial imitation crab sticks (ICSs). Each column represents mean ± SD (*n* = 10). Difference from RC: ** = *p* < 0.01, *** = *p* < 0.001 (Dunnett’s test). PS-RC, premium ICS with real red snow crab (*C. japonicus*) leg meat; PS, premium ICS without real red snow crab leg meat; NS, normal ICS.

**Figure 3 foods-11-01381-f003:**
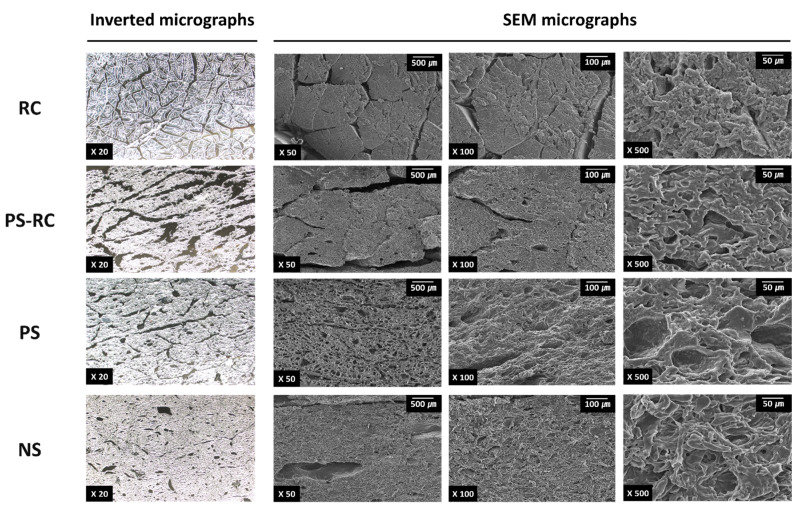
Inverted and SEM micrographs of real snow crab (*C. opilio*) leg meat (RC) and commercial imitation crab sticks (ICSs). Left to right, the magnifications are ×20, ×50, ×100, and ×500. PS-RC, premium ICS with real red snow crab (*C. japonicus*) leg meat; PS, premium ICS without real red snow crab leg meat; NS, normal ICS.

**Figure 4 foods-11-01381-f004:**
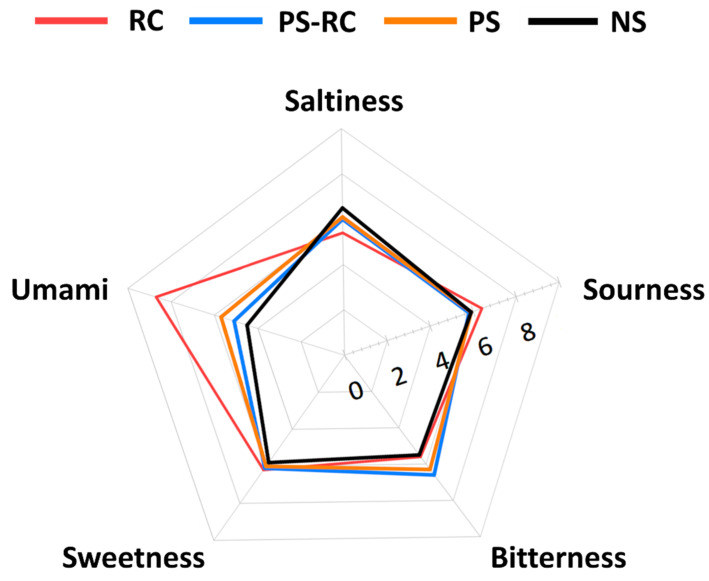
Radar plot for taste data of commercial imitation crab sticks (ICSs) compared to snow crab (*C. opilio*) leg meat (RC). PS-RC, premium ICS with real red snow crab (*C. japonicus*) leg meat; PS, premium ICS without real red snow crab leg meat; NS, normal ICS. Data are shown as means ± standard deviations (*n* = 3).

**Figure 5 foods-11-01381-f005:**
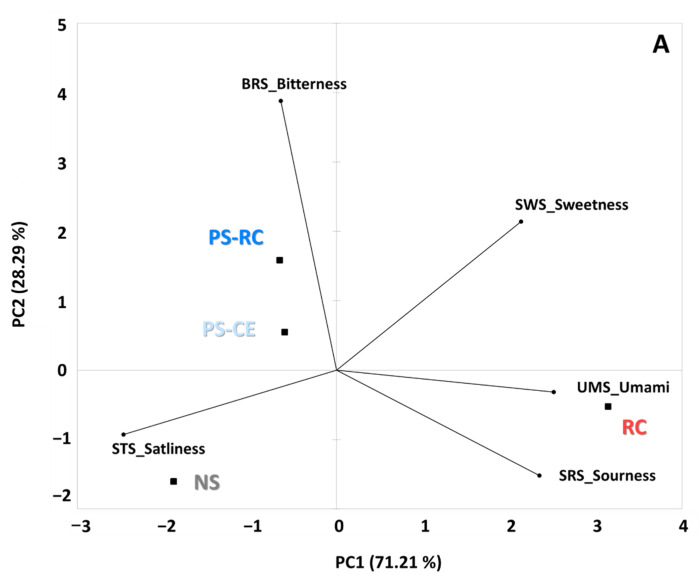
Principle component analysis of real snow crab (*C. opilio*) leg meat (RC) and commercial imitation crab sticks (ICSs) identified via electronic tongue (**A**), electronic nose (**B**), and GC-MS (**C**) testing. PS-RC, premium ICS with real red snow crab (*C. japonicus*) leg meat; PS, premium ICS without real red snow crab leg meat; NS, normal ICS.

**Table 1 foods-11-01381-t001:** Proximate composition of snow crab leg meat and commercial imitation crab sticks.

	RC	PS-RC	PS	NS
**Proximate Composition (g/100 g)**				
Moisture	78.9 ± 0.1 ^d^	73.3 ± 0.2 ^b^	74.6 ± 0.5 ^c^	67.4 ± 0.1 ^a^
Carbohydrate	0.5 ± 0.1 ^a^	12.7 ± 0.1 ^c^	10.7 ± 0.5 ^b^	23.3 ± 0.3 ^d^
Protein	17.8 ± 0.3 ^d^	11.1 ± 0.1 ^c^	10.7 ± 0.2 ^b^	6.3 ± 0.1 ^a^
Lipid	1.2 ± 0.2 ^d^	0.16 ± 0.1 ^a^	0.9 ± 0.1 ^c^	0.8 ± 0.2 ^b^
Ash	1.6 ± 0.1 ^a^	2.8 ± 0.1 ^c^	3.2 ± 0.1 ^d^	2.3 ± 0.1 ^b^

RC, real snow crab (*C. opilio*) leg meat; PS-RC, premium imitation crab stick (ICS) with real red snow crab (*C. japonicus*) leg meat; PS, premium ICS without real red snow crab leg meat; NS, normal ICS. Data are shown as means ± standard deviation (*n* = 3). Multivariate analysis of variance was performed, followed by Duncan’s multiple comparison test. Different letters indicate significant differences (*p* < 0.05).

**Table 2 foods-11-01381-t002:** Hunter color values of snow crab leg meat and commercial imitation crab sticks.

	RC	PS-RC	PS	NS
**Outside**				
*L**	67.2 ± 6.8 ^a^	67.5 ± 3.6 ^a^	73.7 ± 7.0 ^a^	63.5 ± 0.7 ^a^
*a**	10.8 ± 8.5 ^a^	24.6 ± 4.5 ^bc^	17.6 ± 7.0 ^ab^	31.7 ± 2.6 ^c^
*b**	15.4 ± 3.8 ^a^	21.4 ± 4.6 ^a^	21.1 ± 5.1 ^a^	21.2 ± 1.5 ^a^
**Inside**				
Whiteness	78.9 ± 0.8 ^d^	71.1 ± 0.8 ^c^	66.5 ± 2.1 ^b^	58.5 ± 0.7 ^a^

RC, real snow crab (*C. opilio*) leg meat; PS-RC, premium imitation crab stick (ICS) with real red snow crab (*C. japonicus*) leg meat; PS, premium ICS without real red snow crab leg meat; NS, normal ICS; *L**, lightness; *a**, redness; *b** yellowness. Data are shown as means ± standard deviations (*n* = 3). Different letters indicate significant differences (*p* < 0.05) in Duncan’s multiple range test.

**Table 3 foods-11-01381-t003:** Total sugars and nucleotides of snow crab leg meat and commercial imitation crab sticks.

	RC	PS-RC	PS	NS
**Total Sugar (g/100 g)**				
Sucrose	ND	2.5 ± 0.1	4.4 ± 0.1	2.4 ± 0.1
Maltose	ND	ND	ND	ND
Glucose	ND	ND	ND	ND
Fructose	ND	ND	ND	ND
Lactose	ND	ND	ND	ND
**Nucleotides (g/100 g)**				
AMP	24.0 ± 1.4	ND	ND	ND
IMP	ND	11.9 ± 0.2	6.9 ± 0.1	1.6 ± 0.1
GMP	ND	5.1 ± 0.1	2.6 ± 0.1	ND

RC, real snow crab (*C. opilio*) leg meat; PS-RC, premium imitation crab stick (ICS) with real red snow crab (*C. japonicus*) leg meat; PS, premium ICS without real red snow crab leg meat; NS, normal ICS; AMP, adenosine monophosphate; IMP, inosine monophosphate; GMP, guanosine monophosphate; ND, not detected. Data are shown as means ± standard deviations (*n* = 3).

**Table 4 foods-11-01381-t004:** The contents, taste attributes (+ = pleasant, − = unpleasant), taste thresholds and taste activity values of free amino acids in the snow crab leg meat and commercial imitation crab sticks.

Free Amino Acids	Contents (mg/100 g)				Taste Attribute	Taste Threshold (mg/100 mL)	Taste Activity Values			
	RC	PS-RC	PS	NS			PS	PS-RC	PS	NS
Aspartic acid	6.7 ± 0.9 ^a^	0.6 ^b^	1.1 ± 0.2 ^b^	0.5 ^b^	Umami (+)	100	0.1	0.01	0.01	0.01
Glutamic acid	24.2 ± 1.4 ^a^	192.3 ± 23.2 ^ab^	189.1 ± 175.6 ^ab^	257 ± 47.2 ^c^	Umami (+)	30	0.8	6.4	6.3	8.6
Glycine	418.5 ± 99.2 ^a^	408.03 ± 31.2 ^a^	240.9 ± 13.5 ^a^	306.8 ± 155.6 ^a^	Sweet (+)	130	3.2	3.2	1.9	2.4
Alanine	74.9 ± 12.3 ^a^	ND ^b^	4.1 ± 3.6 ^b^	ND ^b^	Sweet (+)	60	1.2	ND	0.1	ND
Threonine	2.6 ± 0.6 ^a^	0.6 ± 0.2 ^b^	0.7 ± 0.1 ^b^	0.3 ^b^	Sweet (+)	260	ND	ND	ND	ND
Serine	5.8 ± 1.6 ^a^	0.5 ± 0.2 ^b^	0.7 ± 0.2 ^b^	0.3 ± 0.1 ^b^	Sweet (+)	150	0.04	ND	ND	ND
Valine	14.2 ± 1.7 ^a^	2.4 ± 1.5 ^b^	1.7 ± 0.2 ^b^	1.9 ± 1.4 ^b^	Sweet/bitter (−)	40	0.4	0.1	0.04	0.1
Proline	52.9 ± 9.6 ^a^	ND ^b^	0.5 ± 0.9 ^b^	ND ^b^	Sweet/bitter (+)	300	0.2	ND	ND	ND
Lysine	4.6 ± 0.4 ^a^	2.4 ± 0.4 ^b^	2.0 ± 0.1 ^b^	0.7 ± 0.2 ^c^	Sweet/bitter (−)	50	0.1	0.1	0.04	0.01
Isoleucine	5.2 ± 1.0 ^a^	0.5 ± 0.1 ^b^	0.5 ± 0.1 ^b^	0.3 ± 0.1 ^b^	Bitter (−)	90	0.1	0.01	0.01-	ND
Leucine	4.5 ± 0.3 ^a^	1.0 ± 0.1 ^b^	1.1 ± 0.1 ^b^	0.7 ± 0.1 ^c^	Bitter (−)	190	0.02	0.01	0.01	ND
Phenylalanine	3.1 ± 0.9 ^a^	0.3 ± 0.5 ^b^	0.2 ± 0.4 ^b^	0.2 ± 0.4 ^b^	Bitter (−)	90	0.03	ND	ND	ND
Histidine	3.8 ± 1.1 ^a^	0.6 ± 0.3 ^b^	ND ^b^	0.7 ± 0.1 ^b^	Bitter (−)	20	0.2	0.03	ND	ND
Arginine	498 ± 34.7 ^a^	1.6 ± 0.6 ^b^	1.1 ± 0.1 ^b^	0.8 ± 0.1 ^b^	Bitter/sweet (+)	50	10.0	0.03	0.02	0.02
Methionine	13.1 ± 7.4 ^a^	0.9 ± 0.3 ^b^	0.8 ± 0.1 ^b^	0.8 ± 0.3 ^b^	Bitter/sweet/sulfurous (−)	30	0.4	0.03	0.03	0.03

RC, real snow crab (*C. opilio*) leg meat; PS-RC, premium imitation crab stick (ICS) with real red snow crab (*C. japonicus*) leg meat; PS, premium ICS without real red snow crab leg meat; NS, normal ICS; ND, not detected. Data are shown as means ± standard deviations (*n* = 3). Different letters indicate significant differences (*p* < 0.05) in Duncan’s multiple range test.

**Table 5 foods-11-01381-t005:** Relative volatile compounds of snow crab leg meat and commercial imitation crab sticks as detected and quantified using an electronic nose.

Compounds	RT^(1)^		RI^(2)^		Sensory Description	RC		PS-RC		PS		NS	
	MXT-5	MXT-1701	MXT-5	MXT-1701		MXT-5	MXT-1701	MXT-5	MXT-1701	MXT-5	MXT-1701	MXT-5	MXT-1701
Butane	14.34	-^(3)^	377	-	Faint	0.07 ± 0.00 ^a^	-	0.12 ± 0.03 ^a^	-	0.12 ± 0.01 ^a^	-	0.10 ± 0.02 ^a^	-
Ethanol	17.80	-	444	-	Alcoholic, pungent, sweet	1.01 ± 0.06 ^c^	-	90.41 ± 8.61 ^b^	-	95.81 ± 1.88 ^b^	-	315.83 ± 23.47 ^a^	-
Pentane	19.34	-	474	-	Alkane, gasoline	0.59 ± 0.26 ^c^	-	4.77 ± 0.25 ^a^	-	2.05 ± 0.39 ^b^	-	2.82 ± 0.36 ^b^	-
2-Propanol	20.76	-	502	-	Acetone, alcoholic, pleasant	2.52 ± 0.13 ^c^	-	82.83 ± 16.31 ^a^	-	18.20 ± 4.34 ^bc^	-	45.13 ± 25.78 ^ab^	-
1-Propanol	23.60	-	556	-	Alcoholic, fruity, pungent	0.18 ± 0.04 ^c^	-	2.12 ± 0.13 ^a^	-	1.63 ± 0.18 ^b^	-	1.38 ± 0.07 ^b^	-
2-methylfuran	26.00	-	601	-	Acetone, burnt, chocolate, gassy	2.67 ± 0.08 ^b^	-	4.91 ± 0.28 ^ab^	-	9.17 ± 3.98 ^a^	-	1.89 ± 0.28 ^b^	-
Trichloroethane	33.26	-	665	-	Chloroform, ethereal, mild, sweet	0.12 ± 0.01 ^c^	-	0.94 ± 0.07 ^a^	-	0.65 ± 0.13 ^ab^	-	0.42 ± 0.21 ^bc^	-
Pyridine	44.82	-	750	-	Amine, burnt, fishy, pungent	0.29 ± 0.05 ^b^	-	0.62 ± 0.08 ^a^	-	0.80 ± 0.11 ^a^	-	0.55 ± 0.19 ^ab^	-
1-Chloropentane	48.14	-	772	-	Green plant, sweet	0.09 ± 0.00 ^d^	-	0.93 ± 0.09 ^b^	-	1.57 ± 0.18 ^a^	-	0.36 ± 0.04 ^c^	-
2.4-Octadiene	55.15	-	816	-	Glue, warm	0.16 ± 0.06 ^a^	-	0.64 ± 0.11 ^a^	-	0.64 ± 0.21 ^a^	-	0.78 ± 0.67 ^a^	-
Methional	69.87	-	903	-	Baked potato, creamy, tomato, vegetable	0.16 ± 0.02 ^a^	-	0.19 ± 0.04 ^a^	-	0.40 ± 0.01 ^a^	-	1.10 ± 1.35 ^a^	-
3-methyl-3-sulfanylbutanol-1-ol	83.44	-	983	-	Broth, chervil, meaty, sweet, vegetable	0.06 ± 0.01 ^a^	-	0.07 ± 0.00 ^a^	-	0.09 ± 0.00 ^a^	-	0.15 ± 0.08 ^a^	-
2,4-Heptadienal,(E,E)-	89.33	-	1018	-	Aldehydic, cinnamon, fatty, green	0.08 ± 0.00 ^a^	-	0.12 ± 0.04 ^a^	-	0.07 ± 0.01 ^a^	-	0.54 ± 0.78 ^a^	-
Decanal	119.10	126.88	1203	1308	Aldehydic, burnt, citrus, fatty, floral	0.10 ± 0.03 ^b^	0.19 ± 0.02 ^a^	0.18 ± 0.02 ^b^	0.20 ± 0.04 ^a^	0.19 ± 0.09 ^b^	0.21 ± 0.08 ^a^	0.55 ± 0.12 ^a^	0.36 ± 0.29 ^a^
Alpha-Terpinen-7-al	131.18	-	1284	-	Fatty, spicy	0.13 ± 0.01 ^a^	-	0.13 ± 0.03 ^a^	-	0.12 ± 0.02 ^a^	-	0.13 ± 0.02 ^a^	-
Tridecane	133.71	-	1302	-	Alkane, citrus, fruity, hydrocarbon	0.13 ± 0.04 ^a^	-	0.14 ± 0.03 ^a^	-	0.16 ± 0.07 ^a^	-	0.26 ± 0.23 ^a^	-
Alpha-Selinene	163.42	-	1523	-	Amber, orange, pepper	0.23 ± 0.04 ^a^	-	0.24 ± 0.03 ^a^	-	0.20 ± 0.05 ^a^	-	0.23 ± 0.02 ^a^	-
Atraton	189.74	-	1736	-	Aldehydic, fatty, spicy	0.09 ± 0.00 ^a^	-	0.09 ± 0.02 ^a^	-	0.09 ± 0.01 ^a^	-	0.08 ± 0.01 ^a^	-
Perfluorononane	-	14.74	-	407	-	-	0.10 ± 0.01 ^b^	-	0.13 ± 0.02 ^a^	-	0.11 ± 0.00 ^ab^	-	0.11 ± 0.01 ^ab^
Acetaldehyde	-	17.89	-	473	Aldehydic, ethereal, fresh, fruity	-	0.15 ± 0.01 ^d^	-	0.84 ± 0.04 ^c^	-	1.12 ± 0.08 ^b^	-	1.91 ± 0.08 ^a^
Methyl formate	-	18.79	-	491	Agreeable, fruity, plum	-	0.42 ± 0.05 ^c^	-	3.86 ± 0.03 ^a^	-	1.25 ± 0.44 ^b^	-	1.01 ± 0.34 ^bc^
Hexane	-	24.12	-	601	Alkane, ethereal, gasoline	-	2.71 ± 0.11 ^b^	-	6.63 ± 0.33 ^ab^	-	9.82 ± 4.10 ^a^	-	1.93 ± 0.11 ^b^
Methylcyclopentane	-	27.37	-	635	Gasoline	-	0.09 ± 0.02 ^b^	-	4.10 ± 0.30 ^a^	-	4.36 ± 0.83 ^a^	-	0.32 ± 0.10 ^b^
Cyclohexane	-	31.82	-	681	Chloroform	-	0.09 ± 0.01 ^d^	-	8.14 ± 0.83 ^a^	-	5.12 ± 0.57 ^b^	-	2.13 ± 0.12 ^c^
Pyridine,2-pentyl-	-	84.25	-	1029	Fatty, green pepper, mushroom, tallowy	-	0.14 ± 0.01 ^a^	-	0.15 ± 0.01 ^a^	-	0.14 ± 0.02 ^a^	-	0.94 ± 1.38 ^a^
2-Octanone	-	95.56	-	1100	Apple, cheese, fatty, fruity	-	0.08 ± 0.04 ^a^	-	0.25 ± 0.02 ^a^	-	0.15 ± 0.02 ^a^	-	0.23 ± 0.18 ^a^
3-Octanol	-	97.86	-	1114	Citrus, nutty, mushroom, herbaceous	-	0.07 ± 0.01 ^a^	-	0.08 ± 0.01 ^a^	-	0.07 ± 0.03 ^a^	-	0.54 ± 0.82 ^a^
5-Methylfurfural	-	103.64	-	1152	Acidic, almond, burnt sugar, caramelized	-	0.07 ± 0.01 ^a^	-	0.07 ± 0.01 ^a^	-	0.09 ± 0.03 ^a^	-	0.28 ± 0.32 ^a^
2-Butyloctanol	-	114.95	-	1226	-	-	0.14 ± 0.07 ^a^	-	0.17 ± 0.02 ^a^	-	0.22 ± 0.05 ^a^	-	0.35 ± 0.24 ^a^
8-Methyl pentadecane	-	156.99	-	1535	-	-	0.25 ± 0.02 ^a^	-	0.23 ± 0.00 ^a^	-	0.24 ± 0.02 ^a^	-	0.36 ± 0.26 ^a^
Methyl tetradecanoate	-	192.17	-	1820	Coconut, cognac, fatty, oily, orris	-	0.12 ± 0.02 ^a^	-	0.10 ± 0.01 ^a^	-	0.10 ± 0.01 ^a^	-	0.27 ± 0.33 ^a^
Tridecyl propanoate	-	199.15	-	1876	-	-	1.07 ± 0.14 ^a^	-	1.13 ± 0.18 ^a^	-	1.11 ± 0.20 ^a^	-	1.13 ± 0.08 ^a^

RC, real snow crab (*C. opilio*) leg meat; PS-RC, premium imitation crab stick (ICS) with real red snow crab (*C. japonicus*) leg meat; PS, premium ICS without real red snow crab leg meat; NS, normal ICS. Data are shown as means ± standard deviations (*n* = 4). Different letters indicate significant differences (*p* < 0.05) in Duncan’s multiple range test. RT^(1)^: retention time; RI^(2)^: retention indices; -^(3)^: not detected.

**Table 6 foods-11-01381-t006:** Relative volatile compounds of snow crab leg meat and commercial imitation crab sticks as detected and quantified by SPME/GC-MS.

Volatile Compounds	RT^(1)^ (min)	RI^(2)^	Content (μg/100 g)				Odor Intensity	Odor Description	ID^(3)^
			RC	PS-RC	PS	NS			
**Acids (4)**									
Oxalic acid isobutyl-nonyl ester	16.85	1078	0.12 ± 0.18	ND	ND	ND^(4)^	1	Crab stick	MS
Ethyl octanoate	20.82	1212	ND	0.32 ± 0.06	0.11 ± 0.16	3.68 ± 0.43			MS/RI
2-Ethylhexyl-oxalic acid	28.57	1511	ND	ND	0.12 ± 0.17	ND			MS
2,5-Dimethyl benzoic acid	32.45	1678	ND	0.06 ± 0.09	ND	ND			MS
**Hydrocarbons (8)**									
Trimethylamine	3.04	<800	0.21 ± 0.30	ND	ND	ND	1	Fish smell	MS/RI
2,6-Dimethyl-octane	16.27	1061	0.23 ± 0.32	ND	ND	ND			MS
Trimethyldecane	16.60	1071	0.12 ± 0.17	ND	ND	ND			MS
Triethyl-phosphate	18.86	1145	ND	0.17 ± 0.03	ND	ND			MS
2,3,4-Trimethyl-1-butene	25.31	1379	0.29 ± 0.01	ND	ND	ND	3	Real crab	MS
Tetradecane	26.15	1412	ND	ND	ND	0.13 ± 0.01			MS/RI
Eicosane	28.45	1505	ND	ND	ND	3.52 ± 4.52			MS/RI
Pristane	33.19	<1700	ND	ND	ND	0.18 ± 0.26			MS
**Heterocyclic Compound (1)**									
Butylated hydroxytoluene	28.92	1526	0.68 ± 0.97	ND	ND	ND			MS
**Ketones (2)**									
2,6-Dimethyl-4-heptanone	14.04	989	ND	0.34 ± 0.02	ND	ND			MS
Benzophenone	31.7	1645	ND	ND	0.30 ± 0.01	ND			MS
**Total**			1.66 ± 1.33	0.83 ± 0.08	0.60 ± 0.23	7.52 ± 4.34			

RC, real snow crab (*C. opilio*) leg meat; PS-RC, premium ICS with real red snow crab (*C. japonicus*) leg meat; PS, premium ICS without real red snow crab leg meat; NS, normal ICS; RT^(1)^, retention time; RI^(2)^, retention index; ID^(3)^, identification; ND^(4)^, not detected. Data are shown as means ± standard deviations (*n* = 3).

## Data Availability

The date presented in this study are available in article.

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
