# Peer review of "Comparison of Imitation Crab Sticks with Real Snow Crab (Chionoecetes opilio) Leg Meat Based on Physicochemical and Sensory Characteristics"

_foods, 2022, doi:10.3390/foods11101381_

Round 1

Reviewer 1 Report

Comments and Suggestions for Authors

Substantial amount of work have been done for these samples, however the why combination is selected is not well explained. Can the authors provide justification why?

Section 2.4

How was the panel trained? What was the dimension 

Figure 1B, the figure is hard to read, please revise the colours.

If this is DoD - how did the authors measure overall acceptance?

Why no analysis performed for AA?

Statistical analyses are all scattered in the manuscript, please put them together in one section in 2.5.

The authors have attempted series of PCAs, have they considered MFA to merge the results?

What would be interesting is to attempt a driver of acceptance analysis to highlight which components corresponds to acceptibility for consumers.

Author Response

[Apr. 29. 2022] Prof. Dr. Arun K. BhuniaEditor-in-ChiefFoods

Re: foods-1697445

 Dear Editor:                  Enclosed, please find the revised version of our manuscript “Comparison of imitation crab sticks with real snow crab (Chionoecetes opilio) leg meat based on physicochemical and sensory characteristics”. We have carefully read all the valuable feedback from our reviewers and revised the manuscript accordingly. The sections that underwent major changes in the manuscript are highlighted in red. We are very grateful for the suggestions and believe that we have addressed all the points raised by the reviewers; our responses are in the pages that follow. We thank you again for your consideration. Please let me know if there is any additional information we can provide. 

Sincerely,

Seungmok Cho,

Department of Food Science and Technology, Pukyong National University, Busan 48513, Korea

Tel.: +82-51-629-5833

Fax.: +82-51-629-5820

scho@pknu.ac.kr

Reviewer 1

We thank the reviewer for the kind review of our study and for providing a valuable critique.

Comment 1: Substantial amount of work have been done for these samples, however the why combination is selected is not well explained. Can the authors provide justification why?

Answer: The imitation crab sticks (ICSs) with various types are sold; however, ICSs can be divided into normal and premium products. In particular, the premium ICSs are produced closer to the real crab meat by addition of real crab meat or processing technique. Therefore, we chose the normal ICS and premium ICSs (with or without the real crab meat).  According to the reviewer’s comment, we revised Introduction and Figure 1 to help understanding of readers.

       [Classification of imitation crab sticks]

Comment 2: How was the panel trained? What was the dimension.

Answer: Panel is consisting of the students of department of Food Science and Technology, Pukyong National University. Panel is selected by their experience in the sensory analysis of food. The sensory panel were trained during 2weeks. Test was performed at the same time every day (three sessions of 1h) and all evaluations took place under controlled sensory conditions. During the training, feedback on score of sensory attributes was performed to increase their ability to recognize differences from control in coded sample. They were sufficiently trained to learn and evaluate the sensory attributes of imitation crab. Dimension of samples was 1 x 1 x 1 (width x length x height) cm.

Comment 3: Figure 1B, the figure is hard to read, please revise the colours.

Answer: We thank the reviewer and fully agree. According to the comment, we revised the colors in Figure 1B.

Comment 4: If this is DoD - how did the authors measure overall acceptance?

Answer: Overall acceptance was evaluated as a comprehensive concept differently from other sensory attributes. We thought that there would be a difference between evaluating overall quality at the same time and evaluating the similarity for sensory items separately. When evaluating the overall acceptance, panelists were requested to evaluate which sample had high similarity from control in terms of overall quality using a 9-point scale (1 = not different, 9 = extremely different).

Comment 5: Why no analysis performed for AA?

Answer: We referred to a previous report that analyzed the taste components of boiled snow crab. We thought that analysis of free amino acids would be appropriate to analyze flavor components in boiled snow crab and imitation crabs. Free amino acids were performed for analyzing the flavor components characteristic of boiled snow crab and imitation crabs.

Comment 6: Statistical analyses are all scattered in the manuscript, please put them together in one section in 2.5.

Answer: Thank you for your comment. All statistical methods were re-written in the 2.5.

Comment 7: The authors have attempted series of PCAs, have they considered MFA to merge the results?

Answer: Thank you for your comment. Multiple Factor Analysis (MFA) is one of the statistical techniques that takes root in PCA. The PCA plots (score and loading plots) data in the study were based on the pearson’s correlation. PCA is a multivariate statistical method that entails data reconstruction and reduction. PCA generates a set of new orthogonal axes or variables known as principal components (PCs) from the original variables. The data sets presented on the orthogonal axes are uncorrelated with one another, and express much of the total variability in the data set through comparison of only a few PCs.

Comment 8: What would be interesting is to attempt a driver of acceptance analysis to highlight which components corresponds to acceptibility for consumers.

Answer: We thank kind comments of the reviewer. This comment will be a great point for future study on this research field. Once again, I appreciate your insight.

Reviewer 2 Report

Comments and Suggestions for Authors

This study investigates similarities between real snow crab and imitation crabs. The objectives of the study need to be clarified, as well as more information needs to be included in the statistical analysis, results and discussion sections. Other comments can be found below.

  • In section 2.5 the authors only refer one statistical technique, ANOVA, but others were used, so should be included. Also authors make no reference to the level of significance they used. This information must be included in this section. In this section reference should be made to all statistical techniques used. There are no information about ANOVA assumptions or ACP assumptions.

In Section 3.1

  • line 220 the authors refer that  “the difference-from-control test was ….between ICSs and RC. The authors should illustrate which test was utilized.

  • Why the authors included “overall acceptance” in the sensory attributes?

  • The sensory evaluation obtained in the radar plot should be clarified for the readers.

In section 3.2 the authors presented five quantitative variables (moisture , protein,…) and a qualitative variable (commercial imitation crab sticks)  with four groups (PS-RC, RS,…NS), so when we have more than one quantitative variable we should begin by a multivariate analysis of variance followed by ANOVA´s and post hoc test, whenever possible.

. Line 303- Based on what method the authors conclude that “chewiness and gumminess correlated positively”

Line 385 to 385. Based on what method the authors conclude that?

From page 11 to 20 the results need to be clarified, as well as more information needs to be included. The PCA results need to be clarified and more information needs to be included. The results are presented in a confusing and unclear way.

Some tables must be rearranged and the graphical representation should be near the interpretation. Otherwise is difficult for the reader

The conclusions needs to improved.

Ref 2- colocar em itálico e da seguinte forma LWT - Food Science and Technology 

Author Response

[Apr. 29. 2022] Prof. Dr. Arun K. BhuniaEditor-in-ChiefFoods

Re: foods-1697445

 Dear Editor:                  Enclosed, please find the revised version of our manuscript “Comparison of imitation crab sticks with real snow crab (Chionoecetes opilio) leg meat based on physicochemical and sensory characteristics”. We have carefully read all the valuable feedback from our reviewers and revised the manuscript accordingly. The sections that underwent major changes in the manuscript are highlighted in red. We are very grateful for the suggestions and believe that we have addressed all the points raised by the reviewers; our responses are in the pages that follow. We thank you again for your consideration. Please let me know if there is any additional information we can provide. 

Sincerely,

Seungmok Cho,

Department of Food Science and Technology, Pukyong National University, Busan 48513, Korea

Tel.: +82-51-629-5833

Fax.: +82-51-629-5820

scho@pknu.ac.kr

Reviewer 2

General comments: This study investigates similarities between real snow crab and imitation crabs. The objectives of the study need to be clarified, as well as more information needs to be included in the statistical analysis, results and discussion sections. Other comments can be found below.

Answer: We thank the reviewer for the kind review of our study and for providing a valuable critique.

Comment 1: In section 2.5, the authors only refer one statistical technique, ANOVA, but others were used, so should be included. Also authors make no reference to the level of significance they used. This information must be included in this section. In this section reference should be made to all statistical techniques used. There are no information about ANOVA assumptions or ACP assumptions.

Answer: We thank the reviewer and fully agree. According to the comment, we revised section 2.5.

Comment 2: In Section 3.1, line 220 the authors refer that “the difference-from-control test was ….between ICSs and RC. The authors should illustrate which test was utilized.

Answer: We thank the reviewer and fully agree. According to the comment, we revised section 3.1.

Comment 3: Why the authors included “overall acceptance” in the sensory attributes?

Answer: Overall acceptance was evaluated as a comprehensive concept differently from other sensory attributes. Score of overall acceptance was evaluated to find out similarity of overall quality of product between commercial ICSs and RC.

Comment 4: The sensory evaluation obtained in the radar plot should be clarified for the readers.

Answer: We thank the reviewer and fully agree. According to the comment, we revised the colors in Figure 1B.

Comment 5: In section 3.2 the authors presented five quantitative variables (moisture , protein,…) and a qualitative variable (commercial imitation crab sticks)  with four groups (PS-RC, RS,…NS), so when we have more than one quantitative variable we should begin by a multivariate analysis of variance followed by ANOVA´s and post hoc test, whenever possible.

Answer: We thank the reviewer and fully agree. According to the comment, we performed multivariate analysis of variance, followed by Duncan’s multiple comparison test. We revised Table 1.

Comment 6: Line 303- Based on what method the authors conclude that “chewiness and gumminess correlated positively”.

Answer: We thank the reviewer and fully agree. According to the comment, we revised this part.

Comment 7: Line 385 to 385. Based on what method the authors conclude that?

Answer: IMP and GMP were not detected in RC. The taste of PS was more similar to that of RC than PS-RC in the sensory evaluation. PS-RC had higher IMP and GMP content than PS. The taste of NS had lowest similarity to that of RC in sensory evaluation. NS had lower IMP and GMP content than PS. Lower content of IMP and GMP did not indicate high similarity in taste attribute. High content of IMP and GMP did not indicate lower similarity. IMP and GMP was not consistent with the similarity in taste between RC and ICSs. There were no consistent results between IMP and GMP content and similarity of taste.

Comment 8: From page 11 to 20 the results need to be clarified, as well as more information needs to be included. The PCA results need to be clarified and more information needs to be included. The results are presented in a confusing and unclear way?

Answer: Thank you for your comment. PCA plots were re-produced for the better visualization.

Comment 9: Some tables must be rearranged and the graphical representation should be near the interpretation. Otherwise is difficult for the reader.

Answer: Thank you for your comment. Tables and the interpretation are placed close together for the reader.

Comment 10: The conclusions needs to improved.

Answer: Thank you for your comment. We improved the conclusions section.

Round 2

Reviewer 1 Report

Comments and Suggestions for Authors

A MANOVA was done for Table 1 but not stated in Section 2.5. Please add

Table 4. Despite the fact it was mentioned in the previous paper - it is still preferred for the readers of the manuscript to know which AA is sig. different when comparing the 4 products. Since an analysis was done for the e-nose but not this & GC-MS data?

Fig 5 can still benefit from the merging of PCAs using MFAs. It'll be good to know and see which compounds are related to the electronic tongue results.

Author Response

Reviewer 1

We thank the reviewer for the kind review of our study and for providing a valuable critique.

Comment 1: A MANOVA was done for Table 1 but not stated in Section 2.5. Please add

Answer: We thank the reviewer and fully agree. According to the comment, we added this content in section 2.5.

Comment 2: Table 4. Despite the fact it was mentioned in the previous paper - it is still preferred for the readers of the manuscript to know which AA is sig. different when comparing the 4 products. Since an analysis was done for the e-nose but not this & GC-MS data?

Answer: We thank the reviewer and fully agree. According to the comment, we performed Duncan’s multiple range test. Major amino acids of RC were Glycine (sweet), alanine (sweet), proline (sweet/bitter), and arginine (bitter/sweet). Arginine (bitter/sweet) contributes to a pleasant overall preference of crab. Arginine (bitter/sweet) and Glutamic acid (umami) were the major free amino acids in ICSs. Glutamic acid content of ICSs can be explained by the addition of MSG to the crab flavoring. Thank you for your comment. Generally, statistical significances are not required, because GC/MS data had too many variables and their ranges of deviations also large.  

Comment 3: Fig 5 can still benefit from the merging of PCAs using MFAs. It'll be good to know and see which compounds are related to the electronic tongue results.

Answer: Thank you for your comment. The results of the complex and listed electronic nose were shown in a form that can easily understand the correlation between the sample and the components using PCA bi-plot.

Reviewer 2 Report

Comments and Suggestions for Authors

I recommend that this manuscript meed some minor revisions. 

Minor issues:

Table 5: I suggest to put mean ± standard deviation in the same line

Line 477: “.. The PC1 and PC2 plots explain…”. PC1 and PC2 are componentes so the authors should writte

“….The PC1 and PC2 components explain….”

Line 478:”… PC1 plot is composed…” delete plot

Author Response

Reviewer 2

Comment 1: Table 5: I suggest to put mean ± standard deviation in the same line.

Answer: It has been done. Thank you for your comment.

Comment 2: Line 477: “.. The PC1 and PC2 plots explain…”. PC1 and PC2 are components so the authors should write  “….The PC1 and PC2 components explain….”.

Answer:. Thank you for your comment. Authors explained the relationship between PCs and components in the manuscript.

Comment 3: Line 478:”… PC1 plot is composed…” delete plot

Answer: It has been done.
